# Isolation, Identification, and Characterization of Endophytic *Bacillus* from Walnut (*Juglans sigillata*) Root and Its Biocontrol Effects on Walnut Anthracnose

Xiaofei Feng [1], Rong Xu [2], Ning Zhao [1], Dongmei Wang [2], Mengren Cun [1] and Bin Yang [2,*]

[1] College of Life Sciences, Southwest Forestry University, Kunming 650224, China
[2] Key Laboratory of Forest Disaster Warning and Control of Yunnan Province, Southwest Forestry University, Kunming 650224, China
* Correspondence: yangbin48053@swfu.edu.cn

**Abstract:** Anthracnose is a major disease of walnut, which seriously reduces the yield and quality of walnut in Yunnan province. Therefore, it is necessary to explore and find a biological control agent for the prevention and control of anthracnose disease. In this study, an endophytic *Bacillus* WB1, with broad-spectrum antibacterial activity was isolated and screened from healthy walnut roots. The strain WB1 was identified as *Bacillus siamensis* WB1 based on morphological characteristics, physiological and biochemical tests, and 16S rRNA gene sequence analysis. *Bacillus siamensis* WB1 produces siderophores and indole-3-acetic acid and solubilizes inorganic phosphate. The strain WB1 not only showed a significant inhibition effect on fourteen phytopathogens, but also showed obvious inhibition on the spore germination of *Colletotrichum acutatum*. Meanwhile, strain WB1 can code genes for the production of antifungal lipopeptides and generate extracellular hydrolytic enzymes (protease, β-1, 3-glucanase, cellulase, and amylase). In addition, WB1 activated the systemic resistance of the host plant by enhancing the activity of defense enzymes, including phenylalanine ammonia lyase (PAL), peroxidase (POD), and polyphenol oxidase (PPO). The results of greenhouse assays also revealed that *B. siamensis* WB1 can effectively reduce the occurrence and severity of walnut anthracnose disease. These results also indicated that *B. siamensis* WB1 is a potential biocontrol agent for walnut anthracnose.

**Keywords:** walnut; endophytic *Bacillus*; *Colletotrichum acutatum*; biological control; antagonistic activity

## 1. Introduction

Walnuts (*Juglans regia* L.), an economically important oil-bearing crop cultivated in China, is susceptible to various pathogens [1,2], in particular to *Colletotrichum* sp., which causes anthracnose. Anthracnose is detrimental to the leaves of walnut trees, resulting in immature fruits. As planting areas and market demands have increased, focus on this pathogen has intensified [3]. In the Yunnan Province, walnut anthracnose usually occurs between June and August, with walnut trees undergoing more serious damage, such as leaf and fruit abscission. Disease prevalence increases under high temperature and humidity, thereby hindering the prevention and treatment of walnut anthracnose. Currently, the prevention and control of walnut anthracnose depends on the utilization of broad-spectrum fungicides [4,5]. However, these fungicides do not easily degrade and therefore contaminate the environment; in addition, resistance development has been reported [6]. Therefore, there is an urgent need to develop sustainable and environmentally friendly chemicals and methods to protect walnut trees from anthracnose and other pathogens.

*Bacillus* is a genus of ubiquitous microorganisms capable of overcoming drought, heat, low temperatures, and other extreme conditions through the formation of endospores. Thus, *Bacillus* has exhibited adaptability and stability in the environment [7]. The genus *Bacillus* is considered to be an outstanding bio-control agent as it synthesizes lipopeptides,

bacteriocins, polyketides, and other active compounds [8,9]. Various *Bacillus* species, including *B. amyloliquefaciens*, *B. subtilis*, *B. thuringiensis*, *B. cereus*, and *B. velezensis*, have demonstrated great potential in controlling plant diseases [10–12]. In addition, *Bacillus* spp. in the rhizosphere can build a symbiotic association with the host and protect it from salinity, arid, and other harsh environmental stresses [13]. ALKahtani et al. [14] and Ismail et al. [15] revealed that endophytic bacteria in the rhizosphere also showed varying activities to produce auxin (indole-3-acetic acid), hydrolytic enzymes (amylase, cellulase, protease, pectinase, xylanase), and phosphate solubilizing. These bacterial endophytes, including *Bacillus* and *Brevibacillus* strains, are a group of plant growth-promoting rhizobacteria (PGPR), which provide a high potential to stimulate plant growth and increase nutrient uptake in plant species. As an essential component of plant–microbe symbionts, endophytic *Bacillus* spp. plays an important role in growth promotion, stress resistance, and biosynthesis of host plants [16]. Plant root systems are responsible for exchanging substances and energy between plants and their surrounding environment [17]. Certain microorganisms inhabit the root tissues, including endophytic *Bacillus* spp., producing antimicrobial metabolites, which can suppress soil-borne phytopathogens, promote plant growth, and improve crop yields [18]. Zou et al. [19] and Cawoy et al. [20] reported that two types of antibiotics are produced by *Bacillus* spp., lipopeptides and bacteriocins. These antibiotics damage the mycelium structure of pathogenic fungi and exhibit broad-spectrum resistance to phytopathogenic fungi. Furthermore, the germination and growth of pathogenic spores are also affected by the active compounds, weakening plant disease transmission [21]. In addition to antibacterial activity, nutrient competition and plant-induced systemic resistance are other important methods used by *Bacillus* to suppress plant diseases [22]. Plant-induced systemic resistance includes defensive enzyme synthesis, phytohormone signaling pathway modulation, and phenylpropionic acid and flavonoid metabolic modulation [22,23]. Gond et al. [24] and Jain et al. [25] showed that endophytic *Bacillus* spp. activate the plant immune system by influencing jasmonic acid, salicylic acid, and ethylene intracellular signaling pathways, thereby improving plant disease resistance. Furthermore, recent studies have demonstrated that endophytic *Bacillus* spp. produce phytohormones, perform nitrogen fixation, and possess phosphate solubilization abilities [26–28]. Xu et al. [29] and Pérez-Montaño et al. [30] showed that endophytic *Bacillus* spp., which coexist with the root tissues, promote the absorption of nitrogen and mineral elements by the host and increase nutrients in the soil via the production of various hydrolases. Thus, endophytic *Bacillus* is a potential microbial resource for biofertilizer and biopesticide utilization. Although the control efficiency of endophytic *Bacillus* spp. on crops has recently been confirmed [31,32], the prevention and control of *C. acutatum* by root endophytic *Bacillus* spp. is still unknown. Therefore, there is an urgent need to investigate the development and applications of rhizosphere endophytic *Bacillus* spp. isolated from walnut roots.

In our study, endophytic *Bacillus* strains isolated from the healthy roots of walnut trees (*Juglans sigillata* L.) were identified and characterized by morphological, biochemical, and molecular tests. Broad-spectrum disease resistance to phytopathogenic fungi and antibiotic biosynthetic genes, and their plant growth-promoting traits, were also investigated. The main objective of our study was to investigate the potential role of the endophytic *Bacillus* in the biological control of walnut anthracnose.

## 2. Materials and Methods

### 2.1. Isolation of Root Endophytic Strains

Two-year-old, healthy walnut seedlings of "iron walnut" (*Juglans sigillata* L.) were collected from Dali City, Yunnan Province, China (25°37′ N 100°13′ E) and used as test samples in the field. Root tips were excised from walnut root tissues and surface-sterilized with 10% sodium hypochlorite and 75% ethanol, as described by Sobolev et al. [33]. The samples were weighed, approximately 0.5 g, and added to 10.0 mL phosphate-buffered saline solution (PBS: NaCl, 8.0 g; KCl, 0.2 g; $Na_2HPO_4$, 1.42 g; and $KH_2PO_4$, 0.27 g, dissolved

in 1 L sterile $H_2O$, pH 7.4). The supernatant (100 μL) was inoculated onto Luria-Bertani (LB) medium (1.0% tryptone, 0.5% NaCl, 2.0% Agar powder, and 0.3% yeast extract, 1 L sterile $H_2O$) and incubated for two days at $25 \pm 2$ °C. In addition, as controls, 100 μL of the final rinse water was spread onto LB plates to evaluate the effect of surface sterilization. Single bacterial colonies were selected for purification, and the purified strains were stored in LB broth with glycerol (25%) at −80 °C.

## 2.2. Identification and Characterization of Root Endophytic Strains

Key morphological and biochemical characteristics of the isolated colonies were determined according to the methods described by Vos, P. et al. [34]. Potential root endophytic strains were identified further using 16S rRNA gene sequencing. Total genomic DNA of the strains was extracted using a DNA extraction kit (Tsingke, Beijing, China) following the manufacturer's instructions. The 16S rRNA gene was amplified using the forward primer 27-F (5'-AGAGTTTGATCCTGGCTCAG-3') and the reverse primer 1492-R (5'-GGTTACCTTGTTACGACTT-3') [35]. Polymerase chain reaction (PCR) was performed in the final reaction mixture (25 μL) containing 1.00 μL deoxynucleotides, 1.0 μL of each primer, 0.3 μL of *Taq* DNA polymerase (5 U/μL), 2.5 μL of *Taq* buffer (10×), 2.0 μL of 20 ng genomic DNA, and 22.0 μL of deionized water. The PCR thermocycling conditions were as follows: initial denaturation at 98 °C for 2.0 min, followed by 35 cycles of 10 s at 98 °C, 15 s at 55 °C for primer annealing, extension at 55 °C for 15 s, and a final elongation step of 5.0 min at 72 °C [36]. The amplified products were purified and sequenced by Tsingke Biotechnology (Beijing, China). Target gene sequences were submitted to the NCBI GenBank database "http://www.ncbi.nlm.nih.gov/BLAST (accessed on 3 August 2022)"; the accession number was OP132790. Similar nucleotide sequences were selected from the EzBioCloud database "https://www.ezbiocloud.net/tools/ani (accessed on 5 August 2022)". A phylogenetic tree of the 16S rRNA gene sequence was constructed by the neighbor-joining method using MEGA-X software [37].

## 2.3. Plant Growth Promoting Assay

The ability of endophytic strains to solubilize phosphate was evaluated using Pikovskaya medium (PVK) [38]. The colonies were incubated on PVK for five days at 28 °C. A halo zone around the bacterial colony indicated phosphate solubilization. Siderophore production assays were performed as described by Alexander et al. [39]. Isolated strains were inoculated on chrome azurol S (CAS) agar media and incubated for five days at 28 °C. Siderophore production was indicated by an orange halo zone around the colonies. Indole acetic acid (IAA) production was evaluated by incubating the isolated strains in LB media supplemented with L-tryptophan (200 mg/L) at 28 °C on a rotary shaker (180 rpm) for three days. The media was subjected to centrifugation at 8000 rpm for 10 min. One milliliter of the supernatant and 2 mL of Salkowski's reagent (49 mL 35% perchloric acid and 1 mL 0.5 M $FeCl_3 \cdot 6H_2O$) were mixed together and incubated in the dark for 30 min [40]. IAA production was indicated by the appearance of a stable pink color in the media. All experiments were performed in triplicate.

## 2.4. Hydrolytic Enzymes Test

The ability of the isolated strain to produce hydrolytic enzymes was evaluated, wherein four extracellular hydrolases, including protease, β-glucanase, cellulase, and amylase, were tested using the hydrolysis circle method [41]. Firstly, the colony was inoculated on agar plates containing skimmed-milk powder, β-1,3-glucan [42], carboxymethy cellulose (CMC) [43], and soluble starch [44], respectively. All the plates were then cultivated at 28 °C for 96 h, and the diameters of the hydrolysis zones around bacterial colonies were observed.

## 2.5. Pathogenic Fungi and Culture Conditions

*Colletotrichum acutatum* swfu013 (ITS: OP811213, 18S rRNA: OP836298, phylogenetic tree of ITS was provided in Supplementary Material), previously isolated from walnut

leaves, was stored in our laboratory (College of Life Sciences, Southwest Forestry University, Kunming, China). Other phytopathogenic fungi, including *Ascochyta* sp., *Epicoccum nigrum* Kink, *Fusicoccum* sp. (isolated from the walnut fruits); *Pestalotiopsis microspore* (Speg) Batista & Peres, *Phyllosticta juglandis* (DC.) Sacc., *Phomopsis* (Sacc.) Bubak (isolated from the walnut leaves); *Fusarium oxysporum*, *Fusarium. graminearum*, *Fusarium oxysporum* f. sp. Vasinfectum (isolated from the walnut root); *Cytospora chrysosperma* (isolated from the *Pinus armandii* branch); *Venturia nashicola* (isolated from the pear leaves); *Alternaria brassicicola* (isolated from the cabbage leaves); and *Pyricularia grisea* (isolated from the paddy leaves). All the phytopathogenic fungi were cultured on potato dextrose agar (PDA) medium (20% potato, 2.0% glucose, 2.0 % and 1.0 L sterile $H_2O$) and stored at 4 °C in our laboratory.

*2.6. Antifungal Activity of Root Endophytic Strains*

2.6.1. Assessment of the Broad-Spectrum Antagonistic Activity of the Isolated Strains

Fourteen common phytopathogens were selected using the dual-culture method to determine the broad-spectrum antagonistic activities of root endophytic strains [45]. Pathogens were incubated for seven days at 25 °C on PDA medium. The mycelial discs (5.0 mm diameter) of each pathogenic fungus were placed in the center of the PDA plates (9.0 cm diameter) for 48 h, and four bacterial colonies were inoculated equidistant (2.5 cm) from the mycelium discs. Mycelial discs without any bacterial colonies served as blank controls. All plates were replicated three times and cultured for an additional seven days at 25 °C. Inhibition rates of mycelial growth for each phytopathogen were calculated according to the following equation [46]:

$$\text{Percentage inhibition} = [(C - T)/C \times 100)]$$

where C = mycelium diameter of fungus in control and T = mycelium diameter of fungus in treatment.

2.6.2. Antagonistic Activity of the Isolated Strains against *C. acutatum* In Vitro

The inhibitory effects of the isolated strain on *C. acutatum*, mycelial growth, and conidial germination were evaluated. The isolated *Bacillus* strains were cultured in LB media by shaking at 150 rpm for 72 h at 25 °C. The culture was subjected to centrifugation (10,000 rpm for 10 min at 4 °C), and the supernatant was filtered through a 0.22 μm syringe-driven filter. Mycelial discs (5.0 mm diameter) of *C. acutatum* were placed in the center of the PDA plates and incubated for 48 h. Four wells (5.0 mm diameter) were punched equidistant (2.5 cm) from the mycelium discs using a sterile puncher. Culture filtrate (20 μL) was added to each well, and the plate was cultured for seven days at 25 °C. The inhibition rates of mycelial growth were calculated. Mycelial discs containing sterile LB media were used as the controls. Furthermore, *C. acutatum* conidia were dispersed in 0.2% glucose solution to obtain a conidial spore suspension of $10^7$ conidia/mL. Conidial germination inhibition was evaluated further by mixing the culture filtrate with 200 μL conidial suspension with equal volumes of cell-free supernatants and incubated for 6 h at 25 °C. Sterile LB media was used as the control instead of the culture filtrate. Microscopic evaluation was used to determine the conidia germination (Zeiss Primo Star). The germination rate was calculated by the equation: germination rate (%) = number of germinated conidia/total number of conidia × 100. The treated conidia were stained with trypan blue (0.04%, Sigma, USA) for 2 min. The cell mortality rate was determined using the following equation:

Cell mortality rate (%) = number of mortality of conidia/total number of conidia × 100. Treatment was repeated three times, and 100 conidia were counted per repetition [47].

*2.7. Detection of Antibiotic Genes from the Endophytic Strains*

The biosynthetic genes of the antimicrobial compounds were amplified using the primers listed in Table 1. Seven genes related to antimicrobial effects, namely surfactin (*SFB*), bacillomycin (*BMYB*), fengycin (*FEND*), iturin (*ITUB*), bacillaene (*BAE*), bacillibactin (*BAC*), and bacilysin (*BLY*), were evaluated in our study. PCR was performed in a final

volume of 25.0 µL containing 1.0 µL deoxynucleotides, 2.0 µL of each primer, 9.5 µL of double distilled water (DD $H_2O$), and 12.5 µL 1× Rapid *Taq* Master Mix (Tsingke, Beijing, China). The PCR amplification procedure was performed using a thermocycler (Applied Biosystems, USA) using the following conditions: initial denaturation at 98 °C for 2 min, 35 cycles of 10 s at 98 °C, relevant annealing temperature for the relevant primer (Table 1), extension at 72 °C for 10 s, and final elongation for 5 min at 72 °C. The amplicons were resolved by gel electrophoresis using a 1% agarose gel stained with ethidium bromide.

**Table 1.** PCR detection of 7 antibiotic biosynthesis genes from isolated strain. Reproduced or adapted from [48], with permission from Elsevier, 2019.

| Product | Genes | Melting Temp | Primer Sequence(5′→3′) | Size (pb) |
|---|---|---|---|---|
| Surfactin | *SFB* | 50.0 | TTCACACAATTAGAGCT ATATGATGATTGCTCCAG | 338 |
| Bacillomycin | *BMYB* | 55.3 | CGAAACGACGGTATGAAT TCTGCCGTTCCTTATCTC | 371 |
| Iturin | *ITUB* | 55.1 | ATCACCGATTCGATTTCA GCTCGCTCCATATTATTTC | 708 |
| Fengycin | *FEND* | 57.6 | TCAGCCGGTCTGTTGAAG TCCTGCAGAAGGAGAAGT | 231 |
| Bacillaene | *BAE* | 57.6 | CTCCGAAAGACGCAGAAT ACCGACTTTATCCGCTCC | 599 |
| Bacillibactin | *BAC* | 57.6 | ATCTTTATGGCGGCAGTC ATACGGCTTACAGGCGAG | 595 |
| Bacilysin | *BLY* | 58.0 | CGAATGTCATATCCACTTTGC AACCGCATCAGCATAAGGA | 429 |

*2.8. Determination of Defense Enzyme Activity with the Walnut Leaf*

Three-year-old walnut trees (*Juglans sigillata* L.) were regularly sprayed with the isolated bacterial suspension (OD600: 0.3; approximately $1 \times 10^8$ CFU/mL) three times daily, for two days. Leaves treated with sterile distilled water were used as the control. The experiment was performed in triplicate, with three walnut trees per replicate. Leaf tissues were collected at one, two, three, four, and five days post inoculation with the isolated strain and stored at −70 °C for further enzyme assays. The activities of the three defense-related enzymes were measured. Phenylalanine ammonia lyase (PAL), peroxidase (POD), and polyphenol oxidase (PPO) activities were evaluated using the method described by Zhou et al. [49]. All enzyme activities were represented as units (U) per gram of fresh weight of samples (FW).

*2.9. The Evaluation of the Biocontrol Efficacy of the Isolated Strains under Greenhouse Conditions*

Assays were performed in a greenhouse at the Southwest Forestry University, Kunming, Yunnan, China, from June to August 2021. Three-year-old walnut seedlings (*Juglans sigillata* L.) with no physical injury or disease were collected, and uniformly sized walnut leaves were cleaned by sterile distilled water and air-dried. A 100 mL *Bacillus* suspension ($1 \times 10^8$ CFU/mL) was used for each treatment. The leaf samples were sprayed, until runoff, three times daily for two days. After treatment with the endophytic strain, the walnut leaves were injured using an aseptic needle to form four symmetrical holes. Then, 20 µL of a conidial suspension of *C. acutatum* was inoculated into the wounds, with sterile distilled water being the control. The inoculated leaves were maintained at 25 °C, 70–80% humidity, under natural light conditions in a greenhouse. Three independent replicates were performed, and 10 leaves were used for each replicate [50]. Disease percentage (%) for each treatment was calculated. The severity of anthracnose was evaluated using the disease severity index (DSI) and control effect (CE). The DSI was calculated based on the percentage of diseased leaves and evaluated on a scale of zero to nine, as described by Xie et al. [51] and Jiang et al. [52]: zero, no disease symptoms; one, <5% of leaves with disease spots; three, 6–10% of leaves with disease spots; five, 11–25%

of leaves with disease spots; seven, 26–50% of leaves with disease spots; nine, >50% of leaves with disease spots. The DSI, biological CE, and disease percentage of the samples were calculated using the following formulae:

$$\text{Disease severity index (DSI)} = [\textstyle\sum(\text{number of leaves at each scale rating} \times \text{the scale rating})/(\text{total number of leaves} \times \text{highest scale rating})] \times 100$$

$$\text{Control effect (CE\%)} = [(\text{Control disease index} - \text{treatment disease index})/\text{control disease index}] \times 100\%$$

$$\text{Disease percentage (DP\%)} = (\text{Total number of diseased leaves})/(\text{Total number of treatment leaves}) \times 100$$

### 2.10. Statistical Analysis

The report data were analyzed using the statistical package SPSS 22.0 (IBM USA). The contribution and significance of the samples were tested using one-way analysis of variance (ANOVA) followed by Duncan's multiple range test ($p > 0.05$). Each treatment was performed in triplicate.

## 3. Results

### 3.1. Isolation and Selection of Bacillus from the Walnut Root

Based on the characteristics of the colonies three days post inoculation with LB media, two larger colonies were selected from the medium according to the morphological characteristics of the bacillus. No colonies were observed in the control plates, and the different colonies were named WB1 and WB2.

### 3.2. Characterization and Identification of the Selected Endophytic Bacillus Strains

The two endophytic bacteria were evaluated for their antagonistic ability using the dual-culture method. The strain WB1 was selected for further studies based on its excellent biocontrol activities against the fourteen phytopathogens. The strain WB1 was cultured in LB media, and its morphological characteristics were reported. The colonies of WB1 were ivory-white with irregular and wrinkled edges. Physiological and biochemical results are shown in Table 2. The strain WB1 was shown to be an endospore-forming, Gram-positive bacterium. The strain showed negative results with the methyl red test, phenylalanine ammonia lyase, and the catalase test and showed positive results with the hydrolysis of amylum, gelatin, and pectin and the Voges-Proskauer test, oxidase, $NH_3$ production, nitrite reduction, urease, and nitrate reduction. Based on morphological, physiological, and biochemical tests, the strain WB1 was confirmed to belong to the *Bacillus* genus.

**Table 2.** Physiological and biochemical characteristic of endophytic bacteria WB1.

| Properties | Activity | Properties | Activity |
|---|---|---|---|
| Catalase test | + | Phenylalanine ammonia lyase | − |
| Hydrolysis of amylum | + | $NH_3$ production | + |
| Methyl red test | − | Nitrite reduction | + |
| Voges-Proskauer test | + | Urease | + |
| Hydrolysis of gelatin | + | Nitrate reduction | + |
| Oxidase | + | Hydrolysis of Pectin | + |
| Gram' s reaction | + | Spore forming | + |

"+" indicates that the biochemical characteristic is "positive". "−" indicates that biochemical characteristic is "negative".

Plant growth-promoting and hydrolytic enzyme synthesis are important characteristics of *Bacillus*. Our study showed that WB1 grew on solid medium containing $Ca_3(PO_4)_2$

(Table 3) and CAS. Clear dissolution halos were observed around the bacterial colony, which indicated that WB1 could solubilize phosphate (Table 3; Figure 1c) and produce siderophore, respectively (Table 3; Figure 1b). The Salkowski assay also showed that WB1 had the ability to produce IAA (Table 3; Figure 1a). Hydrolase determination showed that WB1 could degrade skimmed-milk powder, β-glucan, carboxymethyl cellulose, and soluble starch, and clear hydrolysis halos were observed around the bacterial colony in different solid media (Figure 2), confirming the ability of WB1 to produce protease, β-1,3-glucanase, cellulase, and amylase.

**Table 3.** Characterization of hydrolytic enzymes test and plant growth promoting traits in vitro.

| Isolate | Growth Promoting Traits | | | Hydrolytic Enzymes Production | | | |
|---|---|---|---|---|---|---|---|
| WB1 | P Solubilization | IAA Production | Siderophores Production | Protease | Glucanase | Amylase | Cellulase |
| | + | + | + | + | + | + | + |

"+" indicates that the growth promoting traits or hydrolytic enzymes production is "positive".

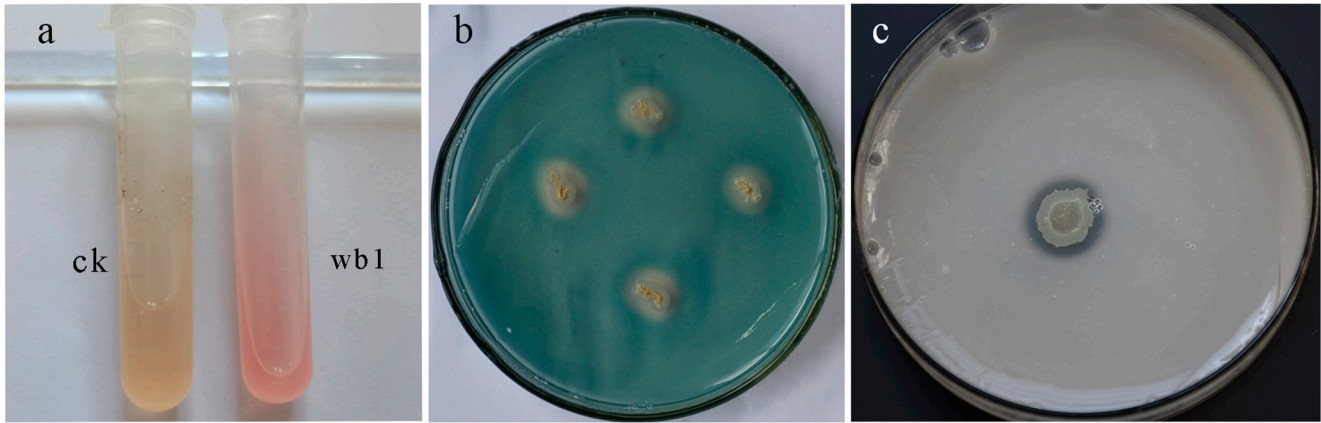

**Figure 1.** Plant growth promoting (PGP) ability of WB1. (**a**) IAA production, pink color was observed in the media of wb1, (**b**) siderophore production by WB1, (**c**) phosphate-solubilizing by WB1.

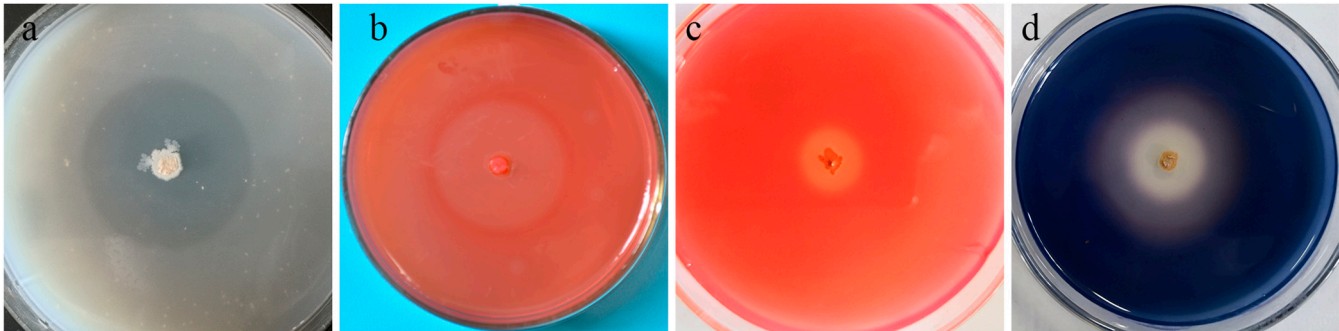

**Figure 2.** Hydrolytic enzymes production ability of WB1. (**a**) protease production, (**b**) β-1,3-glucanase production, (**c**) cellulose production, (**d**) amylase production.

The 16S rRNA gene sequence (1412 bp) of WB1 was submitted to the GenBank database with the accession number OP132790. The phylogenetic tree (Figure 3) showed that WB1 was clustered on the same clade (branch) as *Bacillus siamensis* (AGVF01000043), and the sequence similarity of the 16S rRNA gene was 99.72%. Thus, WB1 was identified as *Bacillus siamensis*.

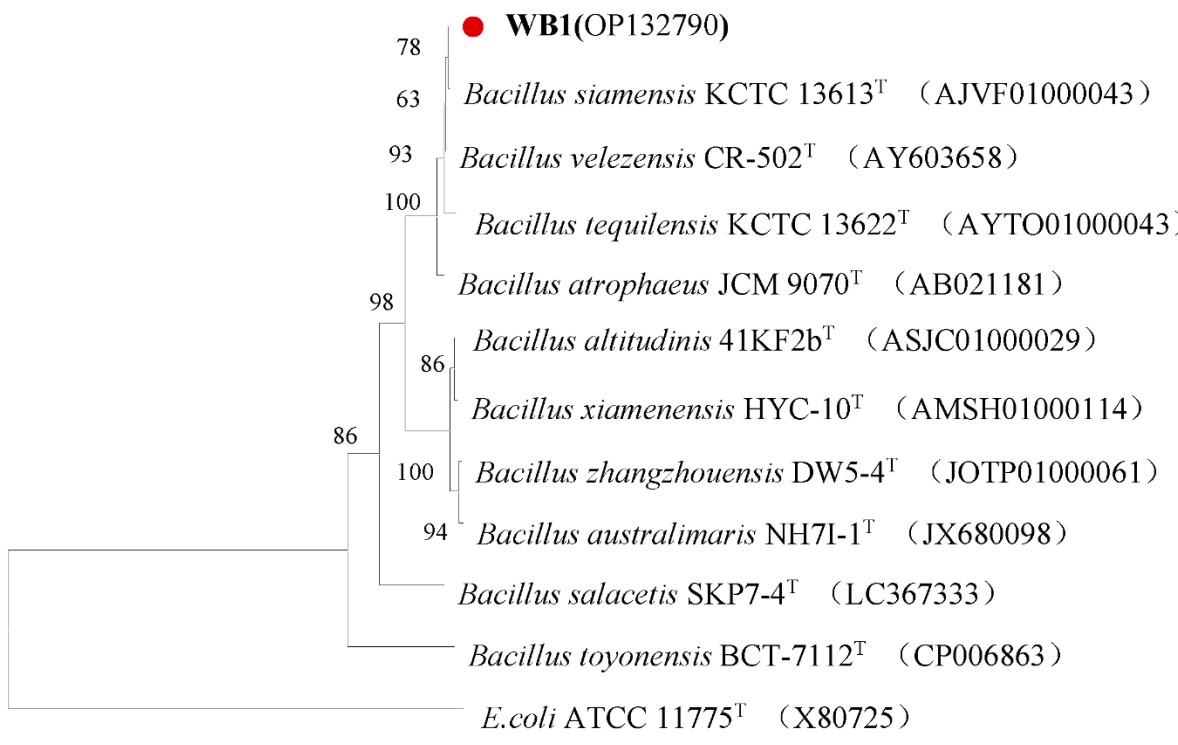

**Figure 3.** Neighbor-Joining phylogenetic tree was constructed by the MEGA-X software based on 16S rRNA. Bootstrap values of 1000 replications are shown next to the branches. The scale bar indicated 0.02 substitutions per nucleotide position.

### 3.3. Antifungal Activity of the Isolated Strains

3.3.1. Antifungal Spectrum of *B. Siamensis* WB1

The inhibition rates and inhibitory zone radius of WB1 on fourteen phytopathogens were calculated, and the results are listed in Table 4. According to these results, *B. siamensis* WB1 inhibited mycelial growth in all the phytopathogens; however, the bacteriostatic effect was different for each pathogen (Table 4; Figure 4). The mycelial growth inhibition rates of WB1 on the selected pathogens ranged from 31.44 ± 5.58% to 60.26 ± 2.90%, with the highest inhibition rate observed for *Cytospora chrysosperma* (60.26 ± 2.90%). The weakest bacteriostatic effect was observed for *Epicoccum nigrum* Kink (31.44 ± 5.58 %). Thus, WB1 is a broad-spectrum antagonistic strain with the ability to inhibit the mycelial growth of several common pathogens.

**Table 4.** Antifungal activity of the WB1 against 14 phytopathogens in a dual plate assay.

| Plant Pathgens | Inhibition Rate (%) | Inhibition Zone Radius (mm) |
|---|---|---|
| *Ascochyta* sp. | 51.48 ± 0.85 [bc] | 7.83 ± 1.07 [bcd] |
| *Epicoccum nigrum* Kink | 31.44 ± 5.58 [c] | 10.88 ± 0.86 [a] |
| *Fusicoccum* sp. | 56.63 ± 3.86 [ab] | 9.17 ± 0.52 [b] |
| *Pestalotiopsis microspore* (Speg) Batista & Peres | 52.15 ± 4.75 [bc] | 3.75 ± 0.30 [h] |
| *Phyllosticta juglandis* (DC.) Sacc. | 33.45 ± 5.74 [c] | 8.78 ± 1.50 [bc] |
| *Phomopsis* (Sacc.) Bubak | 48.98 ± 2.92 [b] | 5.63 ± 1.69 [efg] |
| *Cytospora chrysosperma* | 60.26 ± 2.90 [a] | 12.27 ± 1.12 [a] |
| *F. oxysporum* | 53.33 ± 1.83 [bc] | 6.80 ± 1.21 [def] |

**Table 4.** *Cont.*

| Plant Pathgens | Inhibition Rate (%) | Inhibition Zone Radius (mm) |
| --- | --- | --- |
| *F. graminearum* | 37.16 ± 5.98 [c] | 10.93 ± 0.6 [a] |
| *Venturia nashicola* | 51.93 ± 3.62 [bc] | 6.63 ± 1.03 [def] |
| *Alternaria brassicicola* | 54.26 ± 1.62 [abc] | 7.20 ± 0.63 [cde] |
| *Pyricularia grisea* | 51.59 ± 0.55 [bc] | 5.35 ± 0.74 [fgh] |
| *F. oxysporum* f. *sp. vasinfectum* | 49.44 ± 2.52 [bc] | 4.45 ± 0.72 [gh] |
| *Colletotrichum acutatum* | 46.25 ± 3.21 [bc] | 8.5 ± 1.1 [bc] |

Data are mean ± SD. Different letters (a–h) represent significant differences between groups (Duncan's multiple range test, $p < 0.05$), while the same means insignificant.

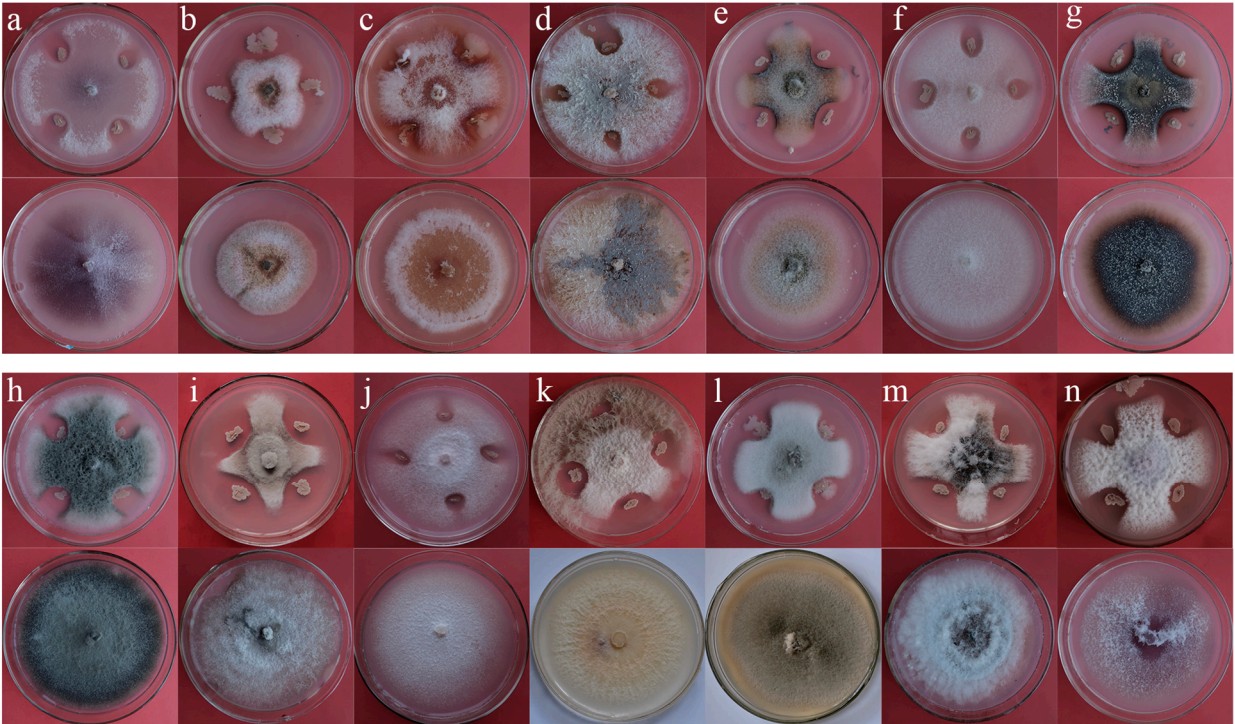

**Figure 4.** Antagonistic of *B. Siamensis* WB1 against 14 phytopathogens: (**a**) *Pyricularia grisea*, (**b**) *F. Graminearum*, (**c**) *Epicoccum nigrum* Kink, (**d**) *Phomopsis* (Sacc.) Bubak, (**e**) *Phyllosticta juglandis* (DC.) Sacc., (**f**) *F. oxysporum*, (**g**) *Cytospora chrysosperma*, (**h**) *Ascochyta* sp., (**i**) *Fusicoccum* sp., (**j**) *F. oxysporum*, f. sp. vasinfectum, (**k**) *Venturia nashicola*, (**l**) *Colletotrichum acutatum*, (**m**) *Pestalotiopsis microspore* (Speg) Batista & Peres, (**n**) *Alternaria brassicicola*).

3.3.2. Antagonistic Activity of *B. Siamensis* WB1 against *C. acutatum* In Vitro

The mycelial growth of *C. acutatum* was significantly suppressed by WB1 supernatant (Figure 5a1). The inhibitory zones and inhibition rates were 8.43 ± 0.18 mm and 36.25 ± 0.83%, respectively. The effects of the culture supernatant on the conidia of *C. acutatum* were also observed. The conidial germination of *C. acutatum* was inhibited by the cell-free supernatants of WB1 (Figure 5a2), with parts of the dead cells stained with trypan blue (Figure 5a3). Spore germination rates 6 h post-treatment were 67.0% and 31.67% for WB1 and the control, respectively, (Table 5). Cell wall staining with trypan blue further showed that the mortality rate in the treated supernatant of WB1 reached 73.33%, but the control had a mortality rate of 3.33% (Table 5).

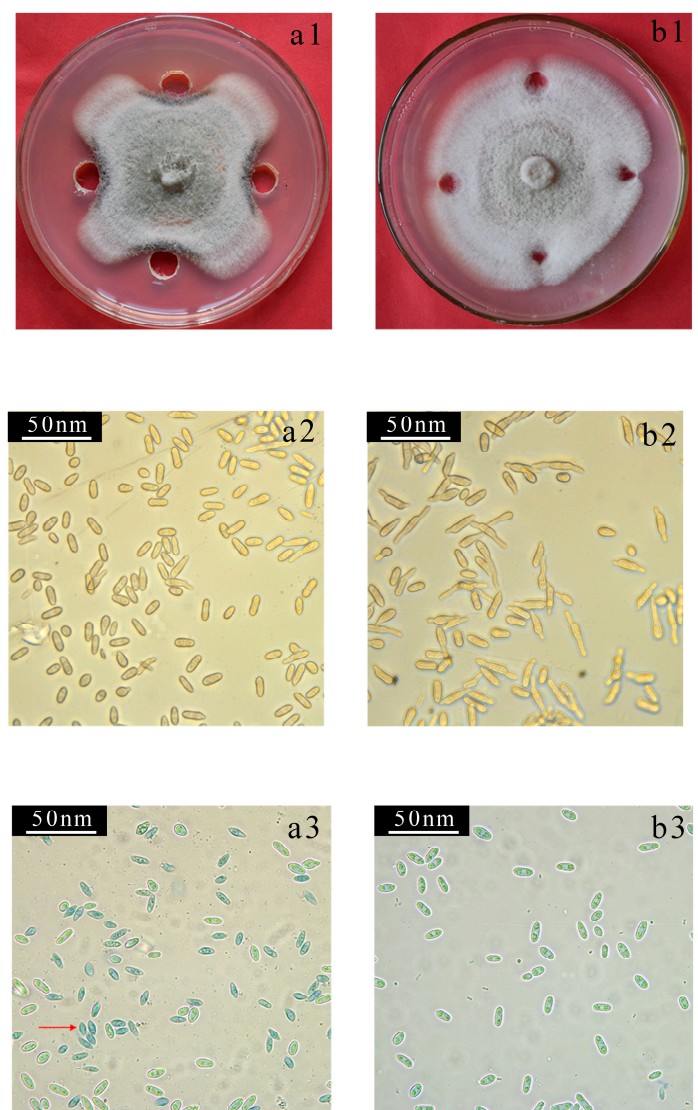

**Figure 5.** The inhibitory effect of the supernatant from *B. siamensis* strain WB1 against the *C. acutatum*. (**a1**) The mycelial growth of *C. acutatum* was suppressed by the supernatant of WB1, (**b1**) control. (**a2**) The conidia germination was suppressed by the supernatant of WB1, (**b2**) control. (**a3**) The conidia mortality rate of *C. acutatum* was treated by the supernatant of WB1; dead cells were stained by Tryphan blue and turn blue, (**b3**) control.

**Table 5.** Inhibitory effect of WB1 fermentation filtrate on the conidia germination of the *C. acutatum*.

| Treatment | Germination Rate (%) | Mortality Rate (%) |
|---|---|---|
| Control | 67.0 ± 1.73 [a] | 3.33 ± 1.53 [a] |
| WB1 | 31.67 ± 3.21 [b] | 73.33 ± 2.89 [b] |

Data are mean ± SD. Different letters (a,b) represent significant differences between groups (Duncan's multiple range test, $p < 0.05$), while the same means insignificant.

### 3.4. Detection of Antibiotic Genes from the Endophytic Strain WB1

Seven antimicrobial compound synthetic genes were selected and amplified by PCR with specific primers. The PCR amplicons are shown in Figure 6 Our study showed that *B. siamensis* WB1 could synthesize surfactin, bacillomycin, fengycin, bacillaene, bacillibactin, and bacilysin, which are considered crucial antimicrobial compounds and play an important role in the broad-spectrum antagonistic activity of WB1.

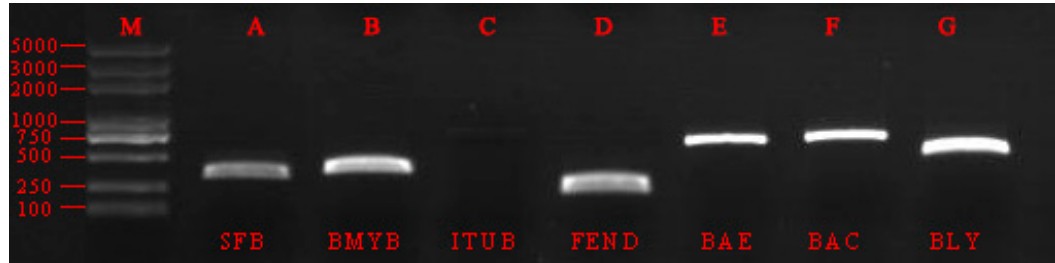

**Figure 6.** Detection of 7 antimicrobial compound synthetic genes amplified in *B. Siamensis* WB1. M: DNA marker, production of A. *SFB*, B. *BMYB*, C. *ITUB*, D. *FEND*, E. *BAE*, F. *BAC*, G. *BLY*.

*3.5. Determination of Defense Enzyme Activity in the Walnut Leaf*

The effect of *B. siamensis* WB1 on walnut trees was evaluated by determining the three different types of resistance-related enzyme activities in walnut leaves. The results indicated that defense-related enzymes in walnut leaves were differentially affected by WB1 (Figure 7). The PAL and PPO activities increased and reached a maximum on the second day of treatment, both of which were significantly ($p < 0.05$) higher than those of the controls. Although PAL and PPO activity declined constantly from day two to day five, PAL and PPO induced by WB1 had significantly enhanced activity compared to the control (Figure 7a,b). Furthermore, POD had the highest activity one day after spraying with the *B. siamensis* WB1 suspension (Figure 7c). Thus, POD in walnut leaves was more sensitive and responded quickly to WB1. Compared to the controls (samples treated with sterile distilled water), POD activity was significantly improved after one day of treatment and stabilized from day three to day five. Although POD activity declined from day three to day five, there was no difference between the experiment and the control. In our study, defense-related enzyme activities, including PAL, POD, and PPO, were induced by root endophytic strains of WB1, which improved resistance in treated walnut leaves and reduced anthracnose damage.

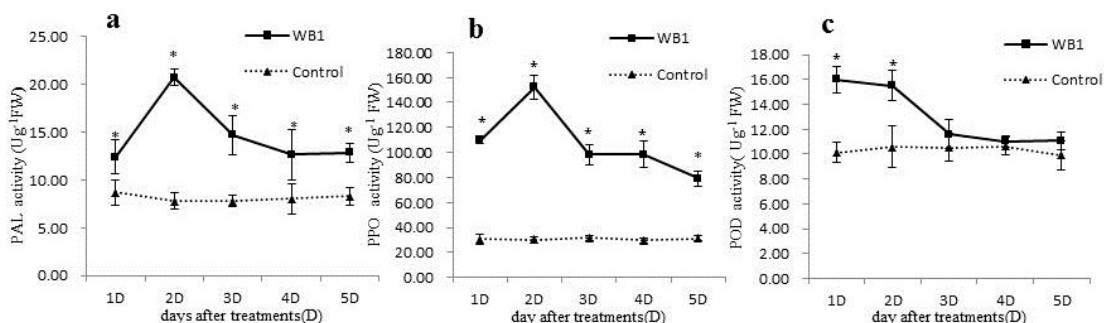

**Figure 7.** The effect of strain WB1 on walnut defense-related enzyme activities. (**a**) PAL, (**b**) PPD, (**c**) POD. The mark "*" above the broken line indicates notable differences between the control and WB1 treatments ($p < 0.05$).

*3.6. Evaluation of the Biocontrol Efficacy of the Isolated Strains under Greenhouse Conditions*

According to the results of the greenhouse experiment, the disease percentage and DSI of walnut anthracnose treated with WB1 were 14.46% and 47.22%, respectively, and they were significantly lower than those of the control plants (Table 6). However, walnut leaf treatments exhibited fewer disease symptoms (Figure 8a,b), and control plants exhibited typical disease symptoms (Figure 8c,d). Therefore, WB1 exhibited good biocontrol efficacy on walnut anthracnose and could be used as a biocontrol agent for controlling walnut anthracnose in the future.

**Table 6.** Control effect of the WB1 on walnut anthracnose in the greenhouse.

| Treatment | Disease Percentage (%) | Disease Severity Index | Control Effect (%) |
|---|---|---|---|
| Control | 58.33 ± 2.06 [a] | 96.29 ± 3.21 [a] | - |
| WB1 | 14.46 ± 4.44 [b] | 47.22 ± 7.35 [b] | 51.32 ± 8.72 |

Data are mean ± SD. Different letters (a,b) represent significant differences between groups (Duncan's multiple range test, $p < 0.05$), while the same means insignificant.

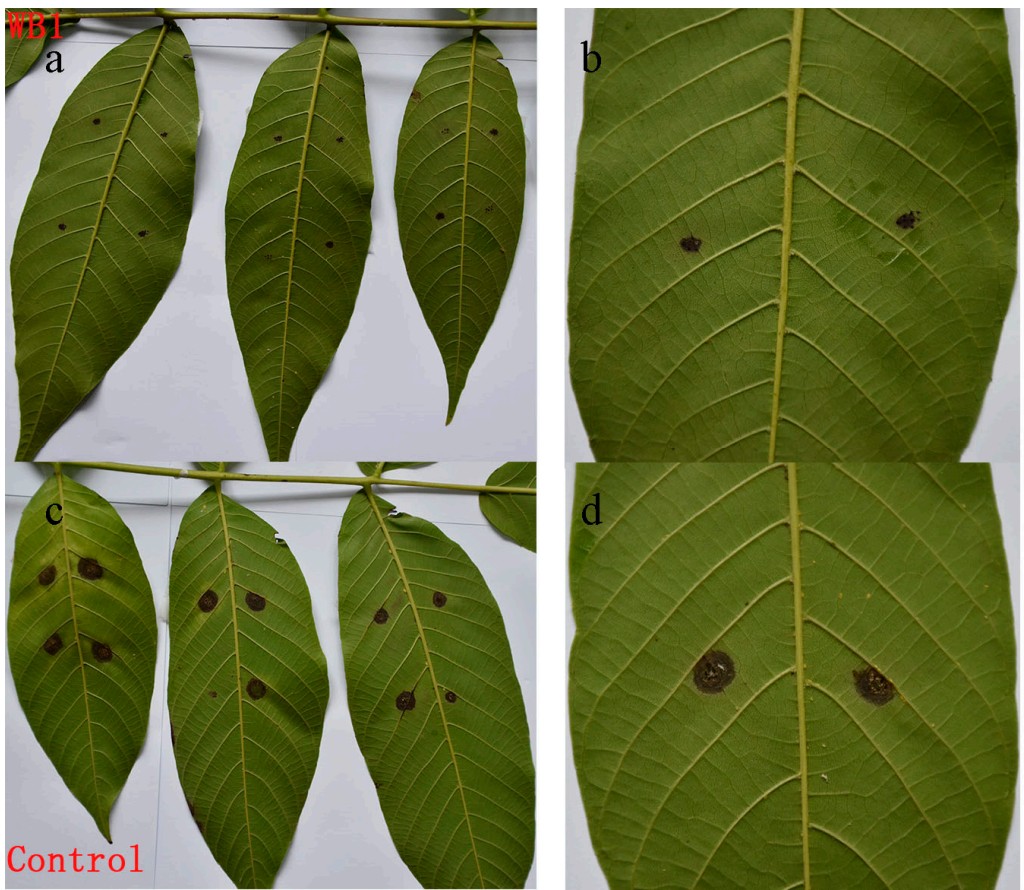

**Figure 8.** The effect of strain WB1 on walnut protective effect on walnut leaves. (**a,b**) The walnut leaves were treated by the supernatant from *B. siamensis* strain WB1, (**c,d**) control.

## 4. Discussion

Walnuts are important economic trees, with walnut cultivation being the main source of income for many families in the poor, remote, mountainous areas of the Yunnan Province. Walnut anthracnose is one of the most devastating diseases, causing severe damage to walnut fruit production and quality, with significant economic losses incurred yearly. Consequently, it is important to develop an environmentally friendly method to protect walnut trees from diseases [53]. The genus *Bacillus*, which plays an important role in biocontrol, particularly the endophytic *Bacillus* spp. with their wide sterilization spectrum, is considered an excellent biocontrol agent providing defense against phytopathogens [54,55]. Endophytic *Bacillus* spp. colonization different plant tissues and, due to long-term systematic evolution, endophytic *Bacillus* has firmly formed a coevolutionary relationship with different hosts [56]. Based on their coexistence with hosts, endophytic *Bacillus* performs beneficial functions in plants without causing adverse effects [57,58]. Vaikuntapu et al. [59] and Babu et al. [60] revealed that rhizosphere endophytic *Bacillus* has beneficial effects on the host via their growth promotion and phytopathogen inhibition. Although several studies have confirmed that the rhizosphere endophytic *Bacillus* spp. have potential for

biological prevention and control, there are only limited reports on endophytic *Bacillus* spp. With biocontrol functions in the walnut rhizosphere. Therefore, our study reported the identification and characterization of endophytic *Bacillus* isolated from walnut roots for the control of *C. acutatum*.

In our study, two endophytic *Bacillus* strains (WB1 and WB2) were isolated from walnut roots using a tissue isolation method. Strain WB1 was selected because of its broad-spectrum antagonistic activity. It exhibited strong inhibitory effects on mycelial growth and the spore germination of *C. acutatum*. A broad antifungal spectrum is an important factor affecting the bacteriostatic efficacy and industrial application of biocontrol agents. Previous investigations have demonstrated that *Bacillus* produces more than one antimicrobial substance to inhibit the growth of multiple phytopathogens. Duan et al. [61] showed that *B. vallismortis* HSB-2 exhibited broad antifungal activity against nine plant fungal pathogens, including *Fusarium* spp., *Alternaria alternata*, *C. acutatum*, *Phoma* sp., *Aspergillus flavus*, *Rhizoctonia solani*, *Penicillium* spp., *Albifimbria verrucaria*, and *Valsa mali*. Endophytic *B. velezensis* HC-8 controlled powdery mildew disease and suppressed the mycelial growth of *Bipolaris maydis*, *F. oxysporum f.* sp. *lycoperisci*, and *F. graminearum* [62]. Jiao et al. [45] also confirmed the production of fengycin, bacillomycin, and other metabolites, which resulted in the endophytic *B.amyloliquefaciens* YN201732 having outstanding antagonistic effects on 12 pathogenic fungi.

Based on morphological and biochemical tests and sequence analysis of the 16S r RNA gene, strain WB1 was identified as *B. siamensis*. *B. siamensis* is considered an important biocontrol agent capable of promoting plant growth and controlling phytopathogens [63]. As an important source of biocontrol microorganisms, *B. siamensis* has been widely used in plant disease control. The endophytic *B. siamensisi* screened from tomato seed has the ability to produce surfactin and bacillomycin, which showed significant antifungal activity and increased the tomato fruit yield [64]. *B. siamensis* H30-3 demonstrated antifungal activities, in vitro, against *Alternaria brassicicola* and *C. higginsianum*, thereby preventing black spot and anthracnose disease in Chinese cabbage [65]. In addition to lipopeptide substances, the volatile organic compounds produced by the *B. siamensis* strain also exhibited significant antifungal activity against *Botrytis cinerea* and could control blueberry postharvest gray mold [66]. Endophytic *B. siamensis* CNE6, which was isolated from chickpea nodules, demonstrated multiple plant growth-promoting properties and significant antagonistic activity against phytopathogenic fungi [67]. Simultaneously, an assay evaluating the plant growth-promoting traits was performed in our study. *B. siamensis* WB1 had the ability to produce siderophores and IAA and could promote phosphate solubilization. As an important phytohormone, IAA can promote tissue growth by accelerating cell division and elongation [68]. Depending on the ability of siderophore production and phosphate solubilization, the soluble mineral content in the soil was improved and soil nutrients were enriched by *B. siamensis* [69]. Thus, our study confirmed that *B. siamensis* WB1 has inherent plant growth-promoting potential.

In vitro antifungal tests demonstrated that *B. siamensis* WB1 has significant antagonistic activity against fourteen phytopathogens, indicating broad-spectrum antagonistic potential. The culture supernatant of WB1 also inhibited mycelial growth and spore germination in *C. acutatum*. According to previous studies, the antibacterial activity of *Bacillus* spp. was attributed to the synthesis of hydrolytic enzymes, production of bacteriostatic compounds, and induction of systemic resistance [70–72]. In our study, WB1 produced protease, β-1,3-glucanase, cellulase, and amylase. Seven defense-related genes were cloned and amplified from WB1 to aid in determining the antifungal mechanisms of WB1. Producing antifungal substances is considered to be the most efficient way of defense by *Bacillus* against phytopathogen invasion. Surfactin, fengycin, and iturin were the main antibiotics of the lipopeptide produced by *Bacillus* spp. [73], which exhibited broad-spectrum antimicrobial activity against pathogens [74,75]. Similarly, the isolated WB1 encoded for lipopeptides and showed mycelial inhibition of multiple phytopathogens. Cell walls play an important role in the cell morphology and defensive capabilities of the host. Cellulose,

chitin, and β-1, 3-glucan were the key components of cell walls with fungal hyphae [76]. Our study revealed that cell wall-degrading enzymes were generated by WB1. Thus, the mycelia of phytopathogens were destroyed by hydrolytic enzymes, which is one of the crucial antifungal mechanisms of WB1 [77]. Furthermore, defense-related enzymes in walnut leaves were enhanced by *B. siamensis* WB1 treatment. Enzymatic activities, including the PAL, PPO, and POD of walnut, were elevated significantly one day after being sprayed with a bacterial suspension of WB1. Defensive enzymes are associated with the systemic-induced resistance of plants. Jiao et al. [61] found that *B. velezensis* HC-8 can improve the antifungal ability of honeysuckle to resist powdery mildew by enhancing the activities of PAL, PPO, and POD. *B. siamensis* also had the ability to induce systemic resistance via regulation of the metabolic and signaling pathways, including phytohormone signaling, phenylpropionic acid, flavonoid metabolism, and the peroxisome pathway [78]. It was confirmed that *B. siamensis* could control plant diseases, including tobacco brown spot disease [51], cucumber *Fusarium* wilt [79], and mango anthracnose [77], due to the defense-related enzymes of plants being strengthened by *B. siamensis*. Our study indicated that *B. siamensis* WB1 triggered systemic resistance activation, which plays an important role in biological control.

The severity of walnut anthracnose was alleviated after treatment with *B. siamensis* WB1 in the greenhouse studies. Compared to the previous studies, *B. siamensis* showed excellent potential for controlling brown spot disease in tobacco [51], sugarcane smut [80], strawberry anthracnose [81], and other plant diseases. The *B. siamensis*-AMU03 was able to protect potato from black scurf due to the production of surfactin, fengycin, and iturin; the lipopeptide compounds inhibit *Rhizoctonia solani* and *Fusarium oxysporum* growth significantly [82]. Thus, the damage caused by *C. acutatum* to walnuts was mitigated by *B. siamensis* WB1, owing to the ability of *B. siamensis* WB1 to produce bacteriostatic compounds, synthesize hydrolytic enzymes, and stimulate systemic resistance in the host.

## 5. Conclusions

In conclusion, the endophytic *B. siamensis* WB1 isolated from walnut roots was regarded as a biocontrol agent with outstanding broad-spectrum antagonistic activity and walnut anthracnose control activities. Our study indicated that the isolated strain *B. siamensis* WB1 showed plant growth-promoting traits, including phosphate solubilization, IAA production, and siderophore production. Due to the production of antifungal lipopeptides and extracellular hydrolytic enzymes, WB1 showed significant antibacterial activity against fourteen phytopathogens, with an obvious negative effect on the spore germination of *C. acutatum*. In addition, WB1 activated systemic resistance by enhancing the activity of defense-related enzymes in host plants, which is a vital factor in walnut anthracnose control. Consequently, based on the multiple beneficial traits of the isolated strain, *B. siamensis* WB1 is a remarkable endophytic *Bacillus* with great potential for application as a biocontrol agent.

**Supplementary Materials:** The following supporting information can be downloaded at: https://www.mdpi.com/article/10.3390/agriculture12122102/s1, Figure S1. The phylogenetic tree of the ITS gene sequence of the *Colletotrichum acutatum* swfu013.

**Author Contributions:** Conceptualization, methodology, and writing—original draft preparation, X.F.; software, validation, data curation, R.X. and N.Z.; formal analysis, resources, D.W. and M.C.; supervision, B.Y. All authors have read and agreed to the published version of the manuscript.

**Funding:** This work was supported by the Natural Science Foundation of Education Department of Yunnan Province (2022J0506), the Industrial Technology Leading Talents Project of Yunnan Province (80201406) and Biological Quality Engineering Project of Yunnan Province (503190106).

**Institutional Review Board Statement:** Not applicable.

**Informed Consent Statement:** Informed consent was obtained from all subjects involved in the study.

**Data Availability Statement:** The 16S rRNA gene sequences (1412 bp) of WB1 have been submitted to the GenBank database with accession number OP132790. The ITS and 18S rRNA genes of Colletotrichum acutatum swfu013 have been sequenced and submitted to the NCBI GenBank database; the accession numbers are OP811213 and OP836298, respectively.

**Conflicts of Interest:** The authors declare no conflict of interest.

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
