# Peer review of "Isolation, Identification, and Characterization of Endophytic Bacillus from Walnut (Juglans sigillata) Root and Its Biocontrol Effects on Walnut Anthracnose"

_agriculture, doi:10.3390/agriculture12122102_

Round 1
Reviewer 1 Report (New Reviewer)
Paper entitled “Isolation, Identification, and Characterization of Endophytic Bacillus From Walnut (Juglans sigillata) Root and Its Biocontrol Effects on Walnut Anthracnose” describes the activity of endophytic bacterial strain B. siamensis to inhibit the growth of different phytopathogens. Also, the antibiotic genes in this strain were identified. The manuscript contains promising data, well written, but needs major revision before acceptance to be published in Agriculture journal for the following reasons.
1- The citation on the first page is not compatible with the title of the manuscript.
2- Line 32, “Sp.” not italic, please revised throughout the manuscript.
3- In line 45, the authors should introduce endophytic bacteria as general and highlight their importance as plant growth-promoting followed by highlighting the endophytic Bacillus spp. I recommend citing the following reference in this part: https://doi.org/10.3390/cells10051059; https://doi.org/10.3390/agronomy10091325; https://doi.org/10.1007/s11816-021-00716-y
4- Lines 56 and 57, this is the identification term of endophytes as generally, not concerned to Bacillus spp., please revised it
5- Line 94, please delete “3.1. Subsection”
6- Subsection 2.3, please mention the source of phytopathogenic fungi and source of isolation, is all isolated from the walnut leaves?
7- Line 260, the authors mentioned that only two endophytic bacteria were isolated from walnut root tips, from my opinion, this is not a logical result because the endophytic bacterial community within plant tissue is high even for the same strain, this finding was confirmed by various literature. Please clarify.
8- In lines 262 – 264, the authors mentioned that “The strain WB1 was selected for further studies based on its excellent biocontrol activities against the fourteen phytopathogens.” Hence these results should be mentioned before the identification to be a logical sequence.
9- Line 269, “gram-positive” should be “Gram-positive”, please revised throughout the manuscript.
10- The sequence of material and method must be matched with the sequence of the result section, for instance, the authors mentioned plant growth promoting activity of bacterial strain after antagonistic activity and detection of antibiotic genes whereas mentioned in the result section before.
11- In GenBank, the bacterial strain was registered as Bacillus sp. whereas in the manuscript the authors identified it as B. siamensis, please clarify and give a reasonable reason.
Author Response
Please see the attachment,Thank you!

Reviewer 2 Report (New Reviewer)
This manuscript titled "Isolation, Identification, and Characterization of Walnut (Ju-glans sigillata) Root Endophytic Bacillus and Its Biocontrol Effects on Walnut Anthracnose" is an interesting study and the study is well deaigned. However, professional English Edit is suggested. In addition, latest literature regarding the subject should be added.
Author Response
Please see the attachment,Thank you!

Reviewer 3 Report (New Reviewer)
Over all study is well designed and nicely compiled.
The writ-up is excellent and very interesting.
I have only one major concern about the identity of the pathogen and correct etiology of the disease.
As in literature different pathogens were reported then C. acutatum such as
1. Wang, Y., Xu, X., Cai, F., Huang, F., Chen, W., & Wang, Q. (2022). First Report of Colletotrichum nymphaeae Causing Walnut Anthracnose in China. Plant Disease, (ja).
2. Ma, T., Yang, C., Cai, F., & Chen, Z. (2022). Morpho-cultural, physiological and molecular characterisation of Colletotrichum nymphaeae causing anthracnose disease of walnut in China. Microbial Pathogenesis, 166, 105537.
3. Zhu, Y. F., Yin, Y. F., Qu, W. W., & Yang, K. Q. (2013, July). Morphological and molecular identification of Colletotrichum gloeosporioides causing walnut anthracnose in Shandong province, China. In VII International Walnut Symposium 1050 (pp. 353-359).
and many others as well.
So my question is for the authors to provide the molecular identity of the pathogen by using at least two marker genes. the sequence should be submitted in the Genbank as well.
And also confirm the koch's postulates of the pathogenicity for the isolated pathogen.
Line 91: Delete an additional full stop sign after investigation.
Some words are marked red in the manuscript. Why it is so?
Author Response
Please see the attachment,Thank you!

Round 2
Reviewer 1 Report (New Reviewer)
Thank you for revised manuscript according to reviewers comments.
The manuscript can be accepted for publication in the current form.
Best regards
Reviewer 2 Report (New Reviewer)
The authors have done the required changes. Therefore the manuscript should be accepted as it is.
Reviewer 3 Report (New Reviewer)
The manuscript is Sufficiently improved and can be considered for further processing.
Thanks
This manuscript is a resubmission of an earlier submission. The following is a list of the peer review reports and author responses from that submission.
Round 1
Reviewer 1 Report
This study deals with the identification, isolation and characterization of an endophytic bacillus from the walnut rhizosphere that exhibits biological control properties toward Colletotrichum gloeosporium. The identified bacillus corresponded to Bacillus siamensis with a broad spectrum of antagonistic activity towards the phytopathogen.
The study on the identified Bacillus siamensis was carried out on a healthy walnut plant root isolate. The same Bacilluss metabolites are expressed in plants infected with Colletotrichum? Is it possible that some characteristics are lost and others exacerbated in the presence of different interactions between microorganisms?
Take care of the wrong punctuation and standardize the writing of the bibliography. It would be convenient to refer to citations of greater accessibility than a Bulletin. These observations are in the manuscript

Author Response
Thank you very much for the reviewers’ comments concerning our manuscript,please see the attachment.

Reviewer 2 Report
see comments in the upload file

Author Response
Thank you very much for the reviewers’ comments concerning our manuscript, Please see the attachment.

Reviewer 3 Report
Minor revisions are required for the publication of this manuscript entitled “Isolation, Identification, and Characterization of Walnut (Juglans sigillata) Rhizosphere Endophytic Bacillus for the Bio-logical Control of Colletotrichum gloeosporium Penz.”. The research conducted by the author is interesting to prevent the growth of phytopathogen using microbes for sustainable agriculture.
1. Line 2: Please improve the title of this manuscript. Walnut rhizophere?? It should be isolation, identification and characterization of endophytic Bacillus from walnut rhizosphere……etc…or rewrite yourself.
2. Improve the abstract language.
3. Improve the introduction part with recent references related to the effect of biofertilizers/PGPR on plant health parameters. Authors may consider these works: https://doi.org/10.1007/s13205-021-02790-z, https://doi.org/10.1007/s13205-020-02561-2, https://doi.org/10.1007/s13205-020-02448-2
4. Please use one-word 16S rDNA or 16S rRNA?
5. Figure 7, on X axis day should be in capital words. Please correct it.
6. Please include the accession number for phytopathogens used in this study.
7. Update the discussion section with recent references.
8. Please follow the same pattern for references.
Author Response

(The authors gave the same response as above.)

Round 2
Reviewer 2 Report
accept in the present form
Author Response
Thank you very much for reviewers’ comments and advice. We have revised the manuscript according to the referees’ comments and the amendments are listed by the attachment.
